# Completion Total Thyroidectomy Is Not Necessary for Papillary Thyroid Microcarcinoma with Occult Central Lymph Node Metastasis: A Long-Term Serial Follow-Up

**DOI:** 10.3390/cancers12103032

**Published:** 2020-10-18

**Authors:** Soon Min Choi, Jin Kyong Kim, Cho Rok Lee, Jandee Lee, Jong Ju Jeong, Kee-Hyun Nam, Woong Youn Chung, Sang-Wook Kang

**Affiliations:** 1Department of Surgery, Severance Hospital, Yonsei Cancer Center, Yonsei University College of Medicine, Seoul 03722, Korea; csm0939@yuhs.ac (S.M.C.); jkkim3986@yuhs.ac (J.K.K.); jandee@yuhs.ac (J.L.); jungjongj@yuhs.ac (J.J.J.); khnam@yuhs.ac (K.-H.N.); woungyounc@yuhs.ac (W.Y.C.); 2Department of Surgery, Yong-In Severance Hospital, Yonsei University College of Medicine, Yong-In 16995, Korea; crlee@yuhs.ac

**Keywords:** papillary thyroid microcarcinoma, completion total thyroidectomy, central lymph node metastasis, recurrence

## Abstract

**Simple Summary:**

The necessity of completion total thyroidectomy is unclear in patients with papillary thyroid microcarcinoma (PTMC) with only pathological central lymph node metastasis (pCLNM). The aim of our study was to determine the necessity of completion total thyroidectomy after an initial surgery by comparing the prognosis according to the presence of pCLNM during a long-term follow-up. We retrospectively compared the pathological central lymph node (pCLN)-positive group of 165 patients who underwent thyroid lobectomy with the pCLN-negative group of 711 patients and found no difference between the two groups in the recurrence rate and disease-free survival rates. Therefore, PTMC patients who underwent thyroid lobectomy with prophylactic central compartment neck dissection and were diagnosed with pCLNM after surgery do not require completion total thyroidectomy.

**Abstract:**

The necessity of completion total thyroidectomy in patients with papillary thyroid microcarcinoma (PTMC) and pathological central lymph node metastasis (pCLNM) who underwent thyroid lobectomy with central compartment neck dissection (CCND) is unclear. We determined the necessity of completion total thyroidectomy by retrospectively comparing the prognosis according to the presence of pCLNM during a long-term follow-up. We enrolled 876 patients with PTMC who underwent thyroid lobectomy with prophylactic CCND from January 1986 to December 2009. Patients were divided according to central lymph node (CLN) metastasis: 165 (18.8%) and 711 (81.2%) in the CLN-positive and CLN-negative groups, respectively. Medical records were reviewed retrospectively, and clinicopathologic characteristics and recurrence rates were analyzed. The CLN-positive group was associated with male sex (*p* = 0.001), larger tumor size (*p* < 0.001), and more microscopic capsular invasion (*p* < 0.001) compared with the CLN-negative group. There was no significant difference between the two groups’ recurrence (*p* = 0.133) or disease-free (*p* = 0.065) survival rates. Univariate and multivariate analyses showed no factors associated with tumor recurrence except male sex (hazard ratio = 3.043, confidence interval 1.117–8.288, *p* = 0.030). Patients who were diagnosed with pCLNM after undergoing thyroid lobectomy with prophylactic CCND do not require completion total thyroidectomy; however, frequent follow-up is necessary for patients with PTMC and pCLNM.

## 1. Introduction

Thyroid cancer (TC) is an increasingly common malignancy, of which papillary thyroid cancer (PTC) is the most common histologic type, with a reported incidence of 80% to 85% [1,2]. Papillary thyroid microcarcinoma (PTMC) is defined as a tumor less than 1 cm in size and has a good prognosis, with a 10-year survival rate over 95% and a recurrence rate of 10% [3,4,5]. Central lymph node (CLN) metastasis has been reported in 20.7–62% of clinically lymph node negative (cN0) PTMC cases [6,7,8,9,10]. Previous studies reported that CLN metastasis was associated with recurrence [11,12,13,14]. Although patients with CLN metastasis have a risk of recurrence, the effectiveness of completion total thyroidectomy in patients diagnosed with pathological CLN metastasis (pCLNM) after surgery is still controversial. According to the 2015 American Thyroid Association (ATA) guidelines, completion total thyroidectomy is necessary in cases of unclear diagnosis after lobectomy, to provide complete resection of multicentric disease, and to allow for efficient radioactive iodine (RAI) therapy; however, the guidelines do not specify pCLNM [15].

Prophylactic central compartment neck dissection (CCND) can affect the pCLNM rate. The mean size and number of metastatic lymph nodes removed during prophylactic CCND were reported as 0.35 cm and 2.6 ± 3 out of 13 ± 5 lymph nodes, respectively [16,17]. As pCLNM was reported to be a micrometastasis, performing prophylactic CCND in patients with cN0 PTMC has been continually under controversy. Multiple studies have compared thyroidectomy alone to thyroidectomy with CCND in cN0 PTMC and reported varying results in complications, recurrence rates, and patient prognosis [18,19,20]. Although opinions vary, most head and neck surgeons and endocrine surgeons in Korea, China, and Japan perform prophylactic CCND even in cases of cN0 PTMC.

Our previous study demonstrated that there was no significant difference in prognosis between CLN-positive and CLN-negative patients with PTMC [21]. As the number of patients with recurrence in the previous study was small, it is necessary to confirm the benefit of completion total thyroidectomy in a CLN-positive group based on more cases and long-term observation. This retrospective study was designed to evaluate the necessity of completion total thyroidectomy in low-risk PTMC patients with pCLNM who underwent thyroid lobectomy with prophylactic CCND at a single medical center.

## 2. Results

### 2.1. Characteristics of the Two Groups

The CLN-positive group showed a greater association with male sex (19.4% vs. 10.4%, *p* = 0.001), a larger tumor size (0.59 ± 0.23 vs. 0.52 ± 0.23, *p* < 0.001), and more microscopic capsular invasion (42.4% vs. 26.6%, *p* < 0.001) than the CLN-negative group. The mean follow-up duration for the CLN-positive and CLN-negative groups was 12.8 ± 4.3 years (range 8.3 to 30.5) and 13.4 ± 4.4 years (range 8.3 to 32.3), respectively. The recurrence rates for the CLN-positive and CLN-negative groups were 7.9% and 4.9%, respectively, but there was no statistically significant difference between the groups. There was no mortality in either the CLN-positive group or the CLN-negative group (Table 1).

### 2.2. Recurrence in the Two Groups

Thirteen patients (7.9%) in the CLN-positive group experienced recurrence during the follow-up period. Six had recurrence in the contralateral lobe, two in both the contralateral lobe and the lateral lymph node (LN), and five in the lateral LN. In the CLN-negative group, 35 (4.9%) patients experienced recurrence. Twenty-seven had recurrence in the contralateral lobe, two in both the contralateral lobe and the lateral LN, and six in the lateral LN. There was no operative bed, central neck, or distant recurrence (Figure 1).

The univariate and multivariate analysis showed that there were no factors associated with tumor recurrence except male sex (hazard ratio [HR] = 3.043, 95% confidence interval [CI] 1.117-8.288, *p* = 0.030) (Table 2). The 10- and 20-year disease-free survival (DFS) rates for the CLN-positive and CLN-negative groups were 99.1% vs. 99.7% and 78.0% vs. 88.9%, respectively (Figure 2). There was no significant difference in DFS rates between the two groups (*p* = 0.065).

When focusing on lateral LN recurrence, the multivariate analysis revealed that it was associated with male sex (HR = 7.113, CI 1.708–29.622, *p* = 0.007), tumor size over 5 mm (HR = 3.790, CI 1.009–14.236, *p* = 0.048), and CLN metastasis (HR = 3.649, CI 1.192–11.169, *p* = 0.023) (Table 3).

### 2.3. Postoperative Complications in the Two Groups

Seven (4.2%) patients experienced postoperative complications in the CLN-positive group and 37 (5.2%) experienced postoperative complications in the CLN-negative group (*p* = 0.519). There was no significant difference between the two groups in postoperative complications, except recurrent laryngeal nerve (RLN) injury. The rate of RLN injury was higher in the CLN-positive group than in the CLN-negative group (1.8% vs. 0.1%, *p* = 0.023) (Table 4).

### 2.4. Comparison Characteristics of the Enrolled Patients in Two Periods

We analyzed the characteristics of the enrolled patients by dividing the period into two parts to compare the older results and recent results: First period, 1986–1997; second period, 1998–2009. The first period consisted of 46 patients. Eight hundred and thirty patients were in the second period. There was no significant difference between the two periods in microscopic capsular invasion, multifocality, CLN metastasis, or complications. Recurrence rate was higher in the first period (19.6% vs. 4.7%, *p* < 0.001), but there was no significant difference between the two periods in mean recurrence-free survival duration (Table 5).

## 3. Discussion

As the incidence of TC increases worldwide, the physicians’ interest in its treatment and prognosis is increasing [22,23]. The long-term recurrence and mortality rates of PTMC are low; thus, there has been constant controversy with regard to the definite treatment guidelines for TC [24,25]. Ito et al. suggested active surveillance (AS) as a therapeutic option for low-risk PTMC [26]. AS can help reduce the risk of surgical complications by avoiding immediate surgery, reducing costs, and improving the quality of life [27,28]. However, the tumor progressed unexpectedly in about 2–15% of patients who were considered suitable for AS, delaying treatment, and these patients’ cancer eventually progressed to an aggressive disease [29,30,31,32,33]. According to previous studies, it was reported that the loco-regional recurrence rate of patients who underwent lobectomy because of PTMC was 2–6%, the distant metastasis rate was 1–2%, and the disease-specific mortality rate was less than 1% [34,35]. Since the clinical characteristics or molecular factors that distinguish low-risk PTMC patients at risk of progression have not been clearly identified, whether surgery or AS has a therapeutic advantage remains controversial.

Since pCLNM had been reported to be relatively high in PTMC, the presence or absence of pCLNM may be estimated as a factor that can affect prognosis and treatment [36,37]. Completion total thyroidectomy was recommended for an unclear diagnosis after lobectomy, complete resection of multicentric disease, and efficient RAI therapy in the 2015 ATA guidelines [15]. However, there is little information on the effectiveness of completion total thyroidectomy in patients who were diagnosed as having pCLNM after thyroid lobectomy.

To demonstrate the usefulness of completion total thyroidectomy in patients diagnosed as having pCLNM after surgery, we followed up low-risk PTMC patients who had undergone thyroid lobectomy with prophylactic CCND for a long time period. During the 13 years of follow-up, 48 patients (5.5%) experienced recurrence. The recurrence rate of the CLN-positive group was higher than that of the CLN-negative group, but there was no significant between-group difference. Our multivariate analysis showed that pCLNM was not associated with recurrence, and DFS was not significantly different between the two groups. Furthermore, there was no distant metastasis or disease-specific mortality. The recurrence rate of PTMC was reported to be 1.96–6% in previous studies. The most common recurrence site was the contralateral lobe, followed by the lateral LN [25,38,39,40,41,42,43]. In our study, 33 patients (33/48, 68.8%) experienced recurrence only in the contralateral lobe and 11 patients (11/48, 22.9%) only in the lateral LN.

Since the most common recurrence site of PTMC was the contralateral lobe, we were faced with a dilemma of whether to perform completion total thyroidectomy. Several studies have reported that 2–7% of patients experience transient recurrent laryngeal nerve (RLN) palsy; 0.5–4.4%, permanent RLN palsy; 7–20%, transient hypoparathyroidism; and 2.5–5.8%, permanent hypoparathyroidism after completion total thyroidectomy [44,45,46,47]. Although the incidence of complications is low, the risk of complications can be completely circumvented by avoiding unnecessary surgery. In addition, because there was no significant difference in long-term prognosis between the CLN-positive and CLN-negative groups and the recurrence rate was very low, we could suggest that completion total thyroidectomy in PTMC with pCLNM was not necessary.

An additional consideration is that pCLNM can affect lateral LN recurrence. Several previous studies have revealed that pCLNM is associated with lateral LN metastasis [37,48,49,50,51]. These studies indicated the possibility of occult lateral LN metastasis in TC with pCLNM. However, there were few studies that proved the relationship between pCLNM and lateral LN recurrence of PTMC after long-term follow up. Ryu et al. reported that the pCLNM rate of TC patients who underwent total thyroidectomy was 33.2%, and lateral LN recurrence was associated with pCLNM [52]. Our analysis showed similar results. Lateral LN recurrence in the CLN-positive group was significantly higher than in the CLN-negative group (odds ratio = 3.893, CI 1.391–10.894, *p* = 0.012). The multivariate analysis found that lateral LN recurrence was associated with male sex, tumor size over 5 mm, and pCLNM. Although pCLNM can develop into lateral LN recurrence, the lateral recurrence rate was very low (7/165, 4.2%). Therefore, active and frequent follow-up is thought to be sufficient for patients with pCLNM. In addition, lateral LN metastasis without pCLNM, so-called skip metastasis, was reported in 6.8–27.8% of PTC patients [48,53,54,55,56,57]. Combining this finding with our results, we carefully question the effectiveness of prophylactic CCND for cN0 PTMC; however, additional prospective randomized studies will be needed in the future.

In several previous studies, the complication rate after thyroidectomy was reported in 5.1% to 26.0% of cases [58,59,60]. Our study showed a relatively low complication rate (44/876, 5.0%). In particular, the rate of hypocalcemia occurrence was 1.3% (11/876) and RLN injury was 0.4% (4/876), which were very low. This result was likely due to the fact that our institution is a high-volume center where specialized endocrine surgeons perform thyroidectomy. Previous studies showing that high volume surgeons had better outcomes relating to complications and prognosis support this [58,59,60,61,62]. Therefore, we can suggest that experienced, specialized endocrine surgeons play a significant part in reducing surgical complications.

Our research showed that the rate of RLN injury was higher in the CLN-positive group. Postoperative complications can occur in any surgery. Since our hospital is an academic institution, trainees frequently participate in surgeries. Although the number of RLN injuries was very small, they did occur. It is difficult to explain why RLN injuries occurred more frequently in the CLN-positive group. Since the surgical range of both groups was determined under the same pre-surgical conditions and indications, we do not think that microscopic CLN metastasis affected the occurrence of surgical complications. According to previous studies, there was no significant difference in complication and clinical outcome when trainees participated in surgery [63,64,65,66]. Therefore, we do not believe that the involvement of trainees influenced the RLN injury rate.

When the patients were divided into two periods and analyzed, the recurrence rate of the patients in the first period was significantly higher. However, this may be the result of the significant difference between the two periods in mean follow-up duration (24.2 ± 2.4 vs. 12.7 ± 3.6, *p* < 0.001). Furthermore, since there was no difference in the mean recurrence-free survival duration, we believe that the difference in recurrence rates between the two periods would not have a significant effect on our study’s results.

There are some limitations to this study. First, as the quality of ultrasonography was not good in the 1980s and 1990s, it could not detect very tiny nodules, which may have been mistaken for recurrence at a later time point. This may have affected the recurrence rate of patients. In our subgroup analysis by period, the fact that the rate of recurrence 5 years after surgery was significantly higher in the first period indicates this possibility (13.0% vs. 2.8%, *p* = 0.003). However, since the number of recurrence patients in the first period was very small, further research is needed. Second, incidental multifocality or microscopic extrathyroidal extension that was undetectable before surgery could have affected the recurrence rates, although multivariate analysis revealed that the two factors were not related to recurrence rates. Third, recent studies have shown that *BRAF*^V600E^ or *TERT* promoter mutations could affect aggressive features and the prognosis of PTC [67,68,69]. However, the effects of *BRAF*^V600E^ or *TERT* promoter mutations on prognosis could not be analyzed in this study because of limited data. Additional research is needed on this in the future.

## 4. Materials and Methods

### 4.1. Ethics Statement

Severance Hospital Institutional Review Board (IRB) gave permission to conduct this study (IRB approval number: 4-2020-0270). Obtaining written informed consent from patients was waived by our ethics committee.

### 4.2. Patients

We retrospectively reviewed the records of 11,336 PTC patients from January 1986 to December 2009. During this period, our institution performed surgeries similar to that described in the 2006 and 2009 ATA guidelines. When the lesion was smaller than 1 cm, unilateral, intrathyroidal, or cN0 thyroid lobectomy with prophylactic CCND was performed, and all other patients underwent bilateral total thyroidectomy with CCND. All patients underwent prophylactic ipsilateral CCND, including the prelaryngeal, pretracheal, and paratracheal LN. Among them, 1209 PTMC patients underwent thyroid lobectomy with prophylactic CCND. On preoperative evaluation, all these patients were diagnosed to not have CLN metastases. In our previous study, 281 patients with PTMC from January 1986 to December 2001 were enrolled. Of them, 185 patients were completely followed up and 691 patients were newly added in this study. Overall, 333 of 1209 patients failed to follow up, and 876 patients were followed up completely. Microscopic capsular invasion was defined as capsule invasion confirmed by a microscope in the final pathologic result. Multifocality was defined as two or more cancer foci in the same lobe. Recurrence was determined when a newly discovered lesion using ultrasonography was confirmed as cancer by a fine needle aspiration biopsy during postoperative follow-up. Clinicopathologic characteristics of the enrolled patients are summarized in Table 6.

Patients were divided into two groups according to the presence of CLN metastasis in the final pathology results: The CLN-positive group and CLN-negative group. Of the 876 patients, 165 (18.8%) were included in the CLN-positive group and 711 (81.2%) in the CLN-negative group. All diagnosed CLN metastases were microscopic metastases. We analyzed the clinicopathologic characteristics and recurrence rates between the two groups.

### 4.3. Statistical Analysis

Statistical analysis was performed using SPSS Statistics for Windows, version 25.0 (IBM Co., Armonk, NY, USA). The Fisher’s exact test and Pearson’s chi-squared test were used for comparing categorical variables. Continuous variables were compared with a Student’s *t*-test. A multivariate Cox proportional hazards regression model was used to evaluate variables for the risk of recurrence. DFS curves were constructed by the Kaplan–Meier method and compared using the log-rank test.

## 5. Conclusions

PTMC patients who underwent thyroid lobectomy with prophylactic CCND and were diagnosed as having pCLNM after surgery did not need to undergo completion total thyroidectomy. Due to the possibility of lateral LN recurrence, active and frequent follow-up is necessary in PTMC with pCLNM. Further research is needed on the effectiveness of prophylactic CCND for cN0 PTMC.

## Figures and Tables

**Figure 1 cancers-12-03032-f001:**
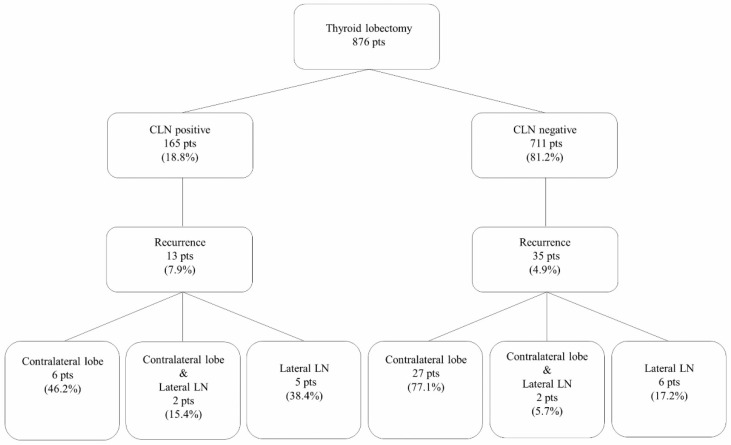
Recurrence in the CLN-positive and CLN-negative groups. Abbreviations: CLN: Central lymph node; LN: Lymph node; pts: Patients.

**Figure 2 cancers-12-03032-f002:**
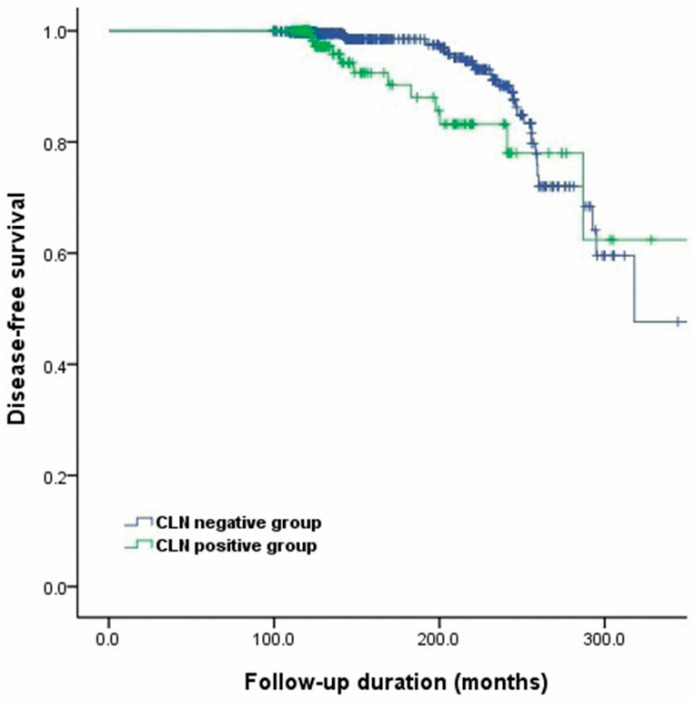
Kaplan–Meier curve for disease-free survival (*p* = 0.065). Abbreviations: CLN: Central lymph node.

**Table 1 cancers-12-03032-t001:** Clinicopathologic characteristics of the central lymph node (CLN)-positive and CLN-negative groups.

Characteristics	CLN-Positive (*n* = 165)	CLN-Negative (*n* = 711)	*p*-Value
Sex			0.001
Female	133 (80.6%)	637 (89.6%)	
Male	32 (19.4%)	74 (10.4%)	
Age (years)			0.355
<55	148 (89.7%)	619 (87.1%)	
≥55	17 (10.3%)	92 (12.9%)	
Tumor size (cm)	0.59 ± 0.23	0.52 ± 0.23	<0.001
Microscopic capsular invasion	70 (42.4%)	189 (26.6%)	<0.001
Multifocality	18 (10.9%)	77 (10.8%)	0.976
Recurrence	13 (7.9%)	35 (4.9%)	0.133
Mean follow-up duration (years)	12.8 ± 4.3	13.4 ± 4.4	0.166

**Table 2 cancers-12-03032-t002:** Cox proportional hazard analysis of variables predicting recurrence after less than total thyroidectomy.

Variable	N	Recurrence	Univariate Analysis		Multivariate Analysis	
			HR (95% CI)	*p*-Value	HR (95% CI)	*p*-Value
Sex				0.016		0.030
Female	770	43 (5.6%)	1.000		1.000	
Male	106	5 (4.7%)	3.307 (1.245–8.787)		3.043 (1.117–8.288)	
Age (years)				0.154		0.194
<55	767	42 (5.5%)	1.000		1.000	
≥55	109	6 (5.5%)	1.887 (0.788–4.523)		1.793 (0.743–4.326)	
Tumor size				0.456		0.732
≤5 mm	505	19 (3.8%)	1.000		1.000	
>5 mm	371	29 (7.8%)	1.254 (0.691–2.276)		1.119 (0.587–2.133)	
Multifocality				0.184		0.243
Absent	781	42 (5.4%)	1.000		1.000	
Present	95	6 (6.3%)	1.844 (0.773–4.400)		1.700 (0.698–4.143)	
Microscopic capsular invasion				0.262		0.540
Absent	617	33 (5.3%)	1.000		1.000	
Present	259	15 (5.8%)	1.432 (0.765–2.680)		1.242 (0.621–2.484)	
CLN metastasis				0.069		0.123
Absent	711	35 (4.9%)	1.000		1.000	
Present	165	13 (7.9%)	1.815 (0.954–3.452)		1.687 (0.868–3.275)	

**Table 3 cancers-12-03032-t003:** Cox proportional hazard analysis of variables predicting recurrence of lateral neck lymph nodes after less than total thyroidectomy.

Variable	N	Lateral LN Recurrence	Univariate Analysis		Multivariate Analysis	
			HR (95% CI)	*p*-Value	HR (95% CI)	*p*-Value
Sex				<0.001		0.007
Female	770	11 (1.4%)	1.000		1.000	
Male	106	4 (3.8%)	10.620 (2.833–39.808)		7.113 (1.708–29.622)	
Age (years)				0.087		0.090
<55	767	12 (1.6%)	1.000		1.000	
≥55	109	3 (2.8%)	3.121 (0.848–11.483)		3.190 (0.835–12.184)	
Tumor size				0.055		0.048
≤5 mm	505	3 (0.6%)	1.000		1.000	
>5 mm	371	12 (3.2%)	3.515 (0.975–12.675)		3.790 (1.009–14.236)	
Multifocality				0.407		0.375
Absent	781	13 (1.7%)	1.000		1.000	
Present	95	2 (2.1%)	1.895 (0.418–8.595)		2.023 (0.427–9.591)	
Microscopic capsular invasion				0.938		0.444
Absent	617	11 (1.8%)	1.000		1.000	
Present	259	4 (1.5%)	1.047 (0.327–3.348)		0.616 (0.178–2.134)	
CLN metastasis				0.007		0.023
Absent	711	8 (1.1%)	1.000		1.000	
Present	165	7 (4.2%)	4.064 (1.463–11.292)		3.649 (1.192–11.169)	

**Table 4 cancers-12-03032-t004:** Postoperative complications of the CLN-positive and CLN-negative groups.

Complications	CLN Positive (*n* = 165)	CLN Negative (*n* = 711)	*p*-Value
Complication			0.519
Absent	158 (95.8%)	674 (94.8%)	
Present	7 (4.2%)	37 (5.2%)	
Hematoma			1.000
Absent	165 (100%)	710 (99.9%)	
Present	0 (0%)	1 (0.1%)	
Seroma			0.331
Absent	164 (99.4%)	696 (97.9%)	
Present	1 (0.6%)	15 (2.1%)	
Hoarseness (transient)			1.000
Absent	163 (98.8%)	701 (98.6%)	
Present	2 (1.2%)	10 (1.4%)	
Hypocalcemia (transient)			0.700
Absent	164 (99.4%)	701 (98.6%)	
Present	1 (0.6%)	10 (1.4%)	
RLN injury			0.023
Absent	162 (98.2%)	710 (99.9%)	
Present	3 (1.8%)	1 (0.1%)	

**Table 5 cancers-12-03032-t005:** Comparison characteristics of the enrolled patients in two periods.

Characteristics	First Period 1986–1997 (*n* = 46)	Second Period 1998–2009 (*n* = 830)	*p*-Value
Sex			0.010
Female	46 (100%)	724 (87.2%)	
Male	0 (0%)	106 (12.8%)	
Age (years)			0.030
<55	45 (97.8%)	722 (87.0%)	
≥55	1 (2.2%)	108 (13.0%)	
Tumor size (cm)	0.63 ± 0.30	0.53 ± 0.23	0.020
Microscopic capsular invasion	10 (21.7%)	249 (30.0%)	0.232
Multifocality	2 (4.3%)	93 (11.2%)	0.219
CLN metastasis	8 (17.4%)	157 (18.9%)	0.797
Complication	1 (2.2%)	43 (5.2%)	0.723
Hematoma	0 (0%)	1 (0.1%)	1.000
Seroma	0 (0%)	16 (1.9%)	1.000
Hoarseness (transient)	0 (0%)	12 (1.4%)	1.000
Hypocalcemia (transient)	1 (2.2%)	10 (1.2%)	0.449
RLN injury	0 (0%)	4 (0.5%)	1.000
Mean follow-up duration (years)	24.2 ± 2.4	12.7 ± 3.6	<0.001
Recurrence	9 (19.6%)	39 (4.7%)	<0.001
Recurrence-free survival duration			
≤5 years	3 (6.5%)	16 (1.9%)	0.073
>5 years	6 (13.0%)	23 (2.8%)	0.003
Mean recurrence-free survival duration (years)	7.4 ± 3.4	5.7 ± 3.6	0.227

**Table 6 cancers-12-03032-t006:** Clinicopathologic characteristics of enrolled patients.

Characteristics	Total Patients (*n* = 876)
Sex	
Female	770 (87.9%)
Male	106 (12.1%)
Age (years)	
<55	767 (87.6%)
≥55	109 (12.4%)
Tumor size (cm)	0.53 ± 0.23
Microscopic capsular invasion	
Absent	617 (70.4%)
Present	259 (29.6%)
Multifocality	
Absent	781 (89.2%)
Present	95 (10.8%)
CLN metastasis	
Absent	711 (81.2%)
Present	165 (18.8%)
Recurrence	
Absent	828 (94.5%)
Present	48 (5.5%)
Mean follow-up duration (years)	13.3 ± 4.4

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
