# Peer review of "Completion Total Thyroidectomy Is Not Necessary for Papillary Thyroid Microcarcinoma with Occult Central Lymph Node Metastasis: A Long-Term Serial Follow-Up"

_cancers, 2020, doi:10.3390/cancers12103032_

Round 1

Reviewer 1 Report

This is an interesting and well-presented study including a large sample-size of the studied population. As the authors state there have been introduced a number of differences in the management of thyroid cancer. My only suggestion to the authors is to compare the results between the older and the most recent years. 

Author Response

Reviewer's comments : This is an interesting and well-presented study including a large sample-size of the studied population. As the authors state there have been introduced a number of differences in the management of thyroid cancer. My only suggestion to the authors is to compare the results between the older and the most recent years.

-> Thanks for reviewing our study and good opinion. As you said, we compared the results between the older and the most recent years by dividing into two periods (1986~1997 and 1998~2009). Based on the results, the additional table (Table 5) has added and more details are described at page 5 paragraph 1.

There was no significant difference between the two periods in microscopic capsular invasion, multifocality, CLN metastasis, or complications. Recurrence rate was higher in the first period (19.6% vs 4.7%, p<0.001), but there was no significant difference between the two periods in mean recurrence-free survival duration.

This result may occur because there was significant difference between two periods in mean follow-up duration (24.2±2.4 vs 12.7±3.6, p<0.001). And since there was no difference in the mean recurrence free survival duration, we believe that the difference in recurrence rates between two periods would not have a significant effect on our study’s results.

Thank you for consideration.

Ward regards,

Soon Min Choi.

Reviewer 2 Report

The authors present an interesting study with clinical impact on surgical treatment of thyroid cancer patients. They retrospectively analyzed 876 patients from 1986-2009 time period divided into central lymph node positive and negative groups and studied the factors related to lymph node status.
Both abstract and summary should include the fact, that it is retrospective study.
Methodology and Results: The pathological characteristics namely microscopic capsular invasion, tumor multifocality and recurrence should be defined in methods part in detail.
Is surgery performed by devoted endocrine surgeons? Or general surgeons? How many thyroidectomies per year?
Why the age was divided under and more than 55 years?
The Tables would be better to incorporate into text so that they are on one page.
Discussion
The aim to perform less total thyroidectomies is outbalanced by post-thyroidectomy complications. Authors can compare in more details their results (Table 4) with other studies and highlight the role of devoted endocrine surgeons to reduce the complications. The fact that trainees are involved in mentioned. Are there any studies on complications in teaching hospital due to trainees? Please discuss in more detail.
Ultrasonography has developed through the study period. Was there any statistical difference in detection rate of metastases and time periods? Discuss both alternatives.

Author Response

Thanks for reviewing our study and good opinion. As you said, we edited our article.

The authors present an interesting study with clinical impact on surgical treatment of thyroid cancer patients. They retrospectively analyzed 876 patients from 1986-2009 time period divided into central lymph node positive and negative groups and studied the factors related to lymph node status.
Both abstract and summary should include the fact, that it is retrospective study.

-> The fact has been added to both abstract and summary. (page 1 line 19, page1 line 28,33)

Methodology and Results: The pathological characteristics namely microscopic capsular invasion, tumor multifocality and recurrence should be defined in methods part in detail.

-> The definition of pathological characteristics was described at page 8 line 234. Microscopic capsular invasion was defined as capsule invasion confirmed by microscope in the final pathologic result. Multifocality was defined as two or more cancer foci in the same lobe. Recurrence was determined when a newly discovered lesion using ultrasonography was confirmed as cancer by a fine needle aspiration biopsy during postoperative follow-up.

Is surgery performed by devoted endocrine surgeons? Or general surgeons? How many thyroidectomies per year?

-> All surgeries were performed by devoted endocrine surgeons. Our data did not include surgeries performed by general surgeons or ENT surgeons. Our institution performs 2400~3000 cases per year. Please see the attachment. The attached file is about our clinical activities of 2019.

Why the age was divided under and more than 55 years?

-> Because AJCC / TNM 8th edition suggested the age standard by 55 years old.

The Tables would be better to incorporate into text so that they are on one page.

-> By reflecting your recommendation, we edited the tables. 

Discussion
The aim to perform less total thyroidectomies is outbalanced by post-thyroidectomy complications. Authors can compare in more details their results (Table 4) with other studies and highlight the role of devoted endocrine surgeons to reduce the complications.

-> We added that content at page 7 paragraph 4 line 183.

Our study showed relatively low complication rate (44/876, 5.0%). In particular, the hypocalcemia was 1.3% (11/876) and the RLN injury was 0.4% (4/876), which were very low. This result was likely due to the fact that our institution is a high-volume center where specialized endocrine surgeons perform thyroidectomy. Previous studies showing that high volume surgeons had better outcomes relating to complications and prognosis support this. Therefore, we can suggest that experienced, specialized endocrine surgeons play a significant part in reducing surgical complications.

The fact that trainees are involved in mentioned. Are there any studies on complications in teaching hospital due to trainees? Please discuss in more detail.

-> We added that content at page 7 paragraph 5 line 197.

According to previous studies, there was no significant difference in complication and clinical outcome when trainees participated in surgery. Therefore, we do not believe that the involvement of trainees influenced the complications.

Ultrasonography has developed through the study period. Was there any statistical difference in detection rate of metastases and time periods? Discuss both alternatives.

-> We compared the results between the older and the most recent years by dividing into two periods (1986~1997 and 1998~2009). Based on the results, the additional table (Table 5) has added and more details are described at page 5 paragraph 1.

The recurrence rate of the patients in the first period was significantly higher. However, this result may occur because there was significant difference between two periods in mean follow-up duration (24.2±2.4 vs 12.7±3.6, p<0.001). And since there was no difference in the mean recurrence free survival duration, we believe that the difference in recurrence rates between two periods would not have a significant effect on our study results.

Recurrence rate after 5 years from surgery was significantly higher in the first period (13.0% vs 2.8%, p=0.003). However, since the number of recurrence patients in first period was very small, it is not clear whether this is due to differences in US quality. All of the above were included at discussion and limitations.

Thank you for consideration.

Ward regards,

Soon Min Choi.
